# Evolutionary Wheat Populations in High-Quality Breadmaking as a Tool to Preserve Agri-Food Biodiversity

**DOI:** 10.3390/foods11040495

**Published:** 2022-02-09

**Authors:** Marco Spaggiari, Mia Marchini, Luca Calani, Rossella Dodi, Giuseppe Di Pede, Margherita Dall’Asta, Francesca Scazzina, Andrea Barbieri, Laura Righetti, Silvia Folloni, Roberto Ranieri, Chiara Dall’Asta, Gianni Galaverna

**Affiliations:** 1Department of Food and Drug, University of Parma, Parco Area delle Scienze 17/A, 43124 Parma, Italy; marco.spaggiari1@studenti.unipr.it (M.S.); luca.calani@unipr.it (L.C.); giuseppe.dipede@unipr.it (G.D.P.); francesca.scazzina@unipr.it (F.S.); laura.righetti@unipr.it (L.R.); chiara.dallasta@unipr.it (C.D.); gianni.galaverna@unipr.it (G.G.); 2Open Fields s.r.l., Strada Madonna dell’Aiuto 7/A, 43126 Parma, Italy; m.marchini@openfields.it (M.M.); r.ranieri@openfields.it (R.R.); 3Department of Veterinary Science, University of Parma, Strada del Taglio 10, 43126 Parma, Italy; rossella.dodi@unipr.it; 4Faculty of Agriculture, Food and Environmental Sciences, Catholic University of the Sacred Heart, via Emilia Parmense 84, 29122 Piacenza, Italy; margherita.dallasta@unicatt.it; 5Molino Grassi SpA, via Emilia ovest 347, Fraore, 43126 Parma, Italy; andreabarbieri@molinograssi.it

**Keywords:** evolutionary populations, sourdough bread, consumer perception, wheat (*Triticum aestivum* L.), bread composition

## Abstract

Plant biodiversity preservation is one of the most important priorities of today’s agriculture. Wheat (*Triticum* spp. L.) is widely cultivated worldwide, mostly under a conventional and monovarietal farming method, leading to progressive biodiversity erosion. On the contrary, the evolutionary population (EP) cultivation technique is characterized by mixing and sowing together as many wheat genotypes as possible to allow the crop to genetically adapt over the years in relation to specific pedoclimatic conditions. The objective of this study was to assess the nutritional, chemical and sensory qualities of three different breads obtained using different organic EP flours, produced following a traditional sourdough process and compared to a commercial wheat cultivar bread. Technological parameters, B-complex vitamins, microelements, dietary fibre and phenolic acids were determined in raw materials and final products. Flours obtained by EPs showed similar characteristics to the commercial wheat cultivar flour. However, significant differences on grain technological quality were found. The breads were comparable with respect to chemical and nutritional qualities. Overall, the sensory panellists rated the tasted breads positively assigning the highest score to those produced with EPs flours (6.75–7.02) as compared to commercial wheat cultivar-produced bread (cv. Bologna, 6.36).

## 1. Introduction

Agriculture, the first actor composing the agri-food system, is currently facing two interconnected crises, such as climate change and biodiversity loss. These challenges are jeopardizing the possibility to provide food manufacturers with high quality raw materials, without affecting price or yield. In conventional agriculture, cereal crops are cultivated repeatedly as monocultures or in rotations that include only two species relying on external inputs such as chemical fertilizers and pesticides [1,2]. On the other hand, ecological principles are followed when considering cereal cultivation in marginal areas, such as mountains, high hills, or organic farming. Among these approaches, the use of a higher inter- and intra-specific diversity and the selection of naturally evolved varieties adapted to the pedoclimatic context over the years [3] are the most efficient ones to ensure cereal yield and quality [4].

On this account, the evolutionary populations (EP) have been introduced with the aim to increase the cultivated biodiversity while ensuring adaptation to the specific pedoclimatic conditions and climate change effect. This concept, introduced more than 60 years ago [5] and now applied by an increasing number of low input farmers [6], relies on the mixing and sowing together of as many genotypes of the same species as possible [7]. At the European Union (EU) level, in 2022 the new organic Regulation came into force describing the rules for certified organic production [8,9]. Regulation EU 218/848 defined new options for reproductive plant material available for organic farmers including evolutionary populations within the organic heterogeneous material (OHM) category. The fact that the seeds of evolutionary populations can now be marketed will most likely increase their availability and their cultivation in the EU.

One of the main concerns regarding EPs-produced-bread is related to the poor technological quality for bakery applications. Indeed, evolutionary breeding has been aimed at improving yield stability under low input agriculture rather than technological properties [10]. However, studies are needed to investigate how EP flour responds to the traditional processing of bread-making. Indeed, bread is recognized as a staple food and a cultural driver, synonymous with symbolic values given its wide and varied preparation methods and recipes. Bread is essentially composed of carbohydrates, like starch, polysaccharides and more complex sugars such as dietary fibres, especially when wholemeal flour is used in dough formulation. Nevertheless, it is a vehicle of other important nutrients belonging to lipids (fatty acids), vitamins (B-group) and bioactive compounds (phenolic compounds). Additionally, the bread formulation method plays an important role on both nutritional and organoleptic characteristics.

Today, sourdough manufacturing is receiving greater attention mainly due to the synergistic effect of specific lactic acid bacteria and yeast strains capable of modifying the whole dough structure and composition, leading to dietary fibre and bioactive solubilization and specific sensory properties [11,12,13,14]. Moreover, sourdough processing is perceived by consumers as a traditional technique which could be considered as added value [15] and a useful tool for a potential whole grain exploitation.

In relation to this, food industry drivers and trends are constantly changing. In fact, consumer food choices are shifting to virtuous producers who consider environmental issues and food system sustainability while designing their food products [16]. Consumers are also starting to pay more attention to the sensory characteristics of food, and their inputs are used by food companies to develop new products [17,18]. The new methodologies developed include CATA (check-all-that-apply) questionnaires, which consist of a lists of words and phrases from which respondents must pick all options they deem relevant [19]. Although a novelty in the fields of sensory and consumer science, these kinds of questionnaires were already being used for vast ranges of products, including bread [19,20,21,22,23]. The latter studies have confirmed that CATA questionnaires are a quick, easy, and dependable way to collect information on consumers’ sensory perceptions when it comes to food and can provide similar information to the time-worn descriptive evaluations by skilled assessors [24].

Given the above, the aim of this study was to (i) study the suitability of organic wheat flours (Type I) obtained from EPs cultivated during the 2016–2017 growing season in the Emilia-Romagna Apennines, Italy, for a traditional sourdough bread-making process; (ii) to analyse the chemical and nutritional profile of the flours and the obtained breads and finally (iii) to assess the consumers’ sensory perception by acceptability and check-all-that-apply (CATA) tests.

## 2. Materials and Methods

### 2.1. Chemicals

Acetonitrile, ethyl acetate, formic acid, acetic acid, methanol (>99.9%) were HPLC-grade, hydrochloric acid (HCl, 37.0%), sodium hydroxide (NaOH, >98.0%), caffeic acid (>98%), p-hydroxybenzoic acid (>99%), p-coumaric acid (>98%), sinapic acid (>98%), gallic acid (>98%) and trans-ferulic acid (>99%), *Folin-Ciocalteu’s* reagent solution were purchased from Sigma-Aldrich (St. Louis, MO, USA). The cis-ferulic acid was obtained by total conversion of a trans-ferulic acid solution under UV light.

### 2.2. Plant Materials

Three bread wheat (*T. aestivum* L.) EPs (Bio2, Grossi and ICARDA) and a modern bread wheat variety (Bologna) were cultivated in a farm located in Vogno di Toano (600 m a.s.l), in the Emilia-Romagna Apennines (Italy) under organic farming, over the 2016–2017 growing season. In October 2016 manure was distributed on the fields and harrowing was performed in order to prepare the soil for sowing. Sowing was performed on 31 October 2016 at a sowing rate of 300 seeds/mq. The seedling emergency date was 5 December 2016 and the harvesting date was 5 July 2017. No treatment was performed for pest control.

The initial nucleus of Bio2 and Grossi EPs consisted of material deriving from long-term conservation, crossbreeding and multiplication activities of local heritage varieties by the Azienda Agraria Sperimentale Stuard (Parma, Italy) and from the Claudio Grossi farm (Parma, Italy), respectively. The local heritage varieties were Ardito, Autonomia B, Carosella, Fiorello, Frassineto, Gentilrosso, Mentana, Terminillo, Verna, Virgilio for Bio2 and Ardito, Virgilio, Miracolo, Gentilrosso, Poulard di Ciano for Grossi.

EP ICARDA was assembled in 2009 by Salvatore Ceccarelli and Stefania Grando with the collaboration of the bread wheat breeders of the International Centre for Agriculture Research in Dry Areas (ICARDA, Beirut, Lebanon) and contained F2, F3 and F4 of 1996 crosses. It arrived in Italy in 2010 thanks to the Italian Association for Organic Agriculture (AIAB, Rome, Italy), in the framework of the EU-FP7 Solibam project. For this, it is also known as Solibam bread wheat EP. In this study, four samples of the ICARDA population were collected from different Italian farms and mixed before sowing. The modern bread wheat Bologna, used as a reference, is a variety by Società Italiana Sementi (SIS, San Lazzaro di Savena, Bologna, Italy).

### 2.3. Cereal Grain Milling and Bread Formulation

#### 2.3.1. Technological Quality Analysis of the Wheat Flours

Test weight and protein content of EPs and Bologna variety were determined using an Infratec 1241 near infrared (NIR) spectrophotometer (FOSS Analytical, Hilleroed, Denmark).

To analyze thousand kernel weight, reading was set at 1000 grains in an optical seed counter (Contador, Pfeuffer, Kitzingen, Germany) and the weight of the grain was measured with a precision balance (SBC 53, Scaltec Instruments, Göttingen, Niedersachsen, Germany).

An aliquot of each wheat grain was milled using a Bona laboratory mill (Labormill, Monza, Italy) and analysed for rheological behaviour following the UNI EN ISO 27971/2008 test method [25] by means of an Alveograph (NG Model, Chopin, Villeneuve-la-Garenne, Cedex France), evaluating the baking strength (W, 10^−4^ J) and the curve configuration ratio (P/L ratio, where P (mm) = dough tenacity; L (mm) = dough extensibility).

#### 2.3.2. Flour Preparation

After appropriate cleaning, the kernels were tempered overnight at room temperature by adding a sufficient amount of water to obtain 16.5% final moisture. The grains were milled into flours using an industrial pilot plant (MLU 202; Bühler, Uzwil, Switzerland) consisting of three breaks (B1 to B3), three reduction (C1 to C3) passages and one laboratory bran duster. Milling fractions from the pilot plant accounted for flour (~65.9% extraction rate, ER), middlings (~10.8% ER) and bran (~17.9% ER). Based on an analysis of the total ash content (American Association of Cereal Chemists, Inc., AACC Method 08-12.01) [26], the flours obtained from all EPs were classified as Type 1 (0.65 < ashes ≤ 0.80 dry basis) while flour from the cv. Bologna was found to be a Type 00 flour (ashes ≤ 0.55 d.b.) in conformity with the Italian standard set out in the Presidential Decree 187/2001 [27]. Therefore, to obtain the same commercial type of flours for all samples, a Type 1 flour was prepared from the Bologna variety by combining 12.27% of middlings (ashes = 2.74% d.b.) with the Type 00 flour obtained (ashes= 0.45% d.b.) according to the equation:(1)(gf∗af )+(gm∗am )=(gm+gf)∗ ax
where *g* is the grams of flour (f) or middlings (m), a is the ash content (%, d.b.) of flour (f), middlings (m) and reconstituted flour (x). After reconstitution, cv. Bologna Type 1 flour (FBo) had an 80% ER against the 66.2%, 64.8% and 64.7% of Bio2 EP Type 1 flour (FB), Grossi EP Type 1 flour (FG) and ICARDA EP Type 1 flour (FI), respectively.

#### 2.3.3. Bread Formulation

Four breads (Bio2, ICARDA, Grossi EPs and cv. Bologna Type 1 flours, Figure 1) were produced twice in a baking laboratory by the same professional baker using sourdough manufacturing process. The recipe was: wheat flour (2500 g), sourdough (750 g, prepared by the professional baker), salt (60 g), malt (45 g), extra virgin olive oil (30 mL) and water (~1250 mL). The sourdough starter, commonly used by the same baker in bread-making, was fed twice with organic bread wheat flour Type 0 (Molino Grassi, Parma, Italy) and left to leaven in a prover under controlled conditions (30 °C, 86% relative humidity, RH) for two days. Small adjustments to the bread-making process were made in terms of leavening time, while the dough’s workability was improved by the baker’s expertise. All breads were prepared by mixing the ingredients in a spiral mixer (SPI 45 F E, Esmach, Vicenza, Italy) for 10 min at low speed and 8 min at high speed. More water was added during the kneading depending on the dough’s workability resulting in the following water additions: 480 mL for bread produced using cv. Bologna (BBo), 440 mL for bread produced using ICARDA and Grossi EPs (BI and BG, respectively), 380 mL for bread produced using Bio2 EP (BB). Bulk fermentation was carried out in a prover (BFM 6080, Climother, Bongard Esmach, Italy) under controlled conditions (28 °C, 86% RH) for 90 min. The fermented dough was then divided into 1 kg loaves and placed back into the prover to rest for 15–30 min. Subsequently, the loaves were put into rattan baking molds, proved (28 °C, 86% RH) for 80 min and baked for 60 min at 215 °C in an electric oven (EMT 4/6040, Tagliavini, Parma, Italy). After baking, the loaves were cooled to room temperature, cut into equal slices and immediately used for the sensory and hedonic analysis, or otherwise lyophilized, homogeneously minced under nitrogen, and kept at −20 °C until extraction and analysis.

### 2.4. Protein, Lipids, Dietary Fibre Components and Carbohydrates of Breads

Fat content was determined by Soxhlet (American association Of Analytical Chemistry international, AOAC 922.06 [28]), using diethyl-ether as solvent. FAs profile was determined according to Dall’Asta et al. [29]. The FAs were identified and the relative percentage, calculated using the area under each peak. Results were also reported as saturated (SFA), monounsaturated (MUFA) and polyunsaturated (PUFA) fatty acids in accordance with their unsaturation degree.

Crude nitrogen content was determined following the Kjeldahl method (AOAC 950.36 [28]) using 5.7 as conversion factor. The analysis of high molecular weight insoluble dietary fibre (HMWIDF), high molecular weight soluble dietary fibre (HMWSDF), low molecular weight soluble dietary fibre (LMWSDF) and total dietary fibre (TDF) content in flours and formulated breads was carried out by an external accredited laboratory of food analysis (UNI CEI EN ISO/IEC 17025:2005 [30], Accredia, Lab. n. 0490), using an official enzymatic-gravimetric method (AOAC 2011.25 2013, [28]). Carbohydrates were determined by difference. Lastly, the determination of resistant starch (RS) was undertaken using the AOAC Method 2002.02 [28] for Resistant starch (Megazyme kit, USA). Results were expressed as g/100 g on dry weight basis.

### 2.5. Determination of Magnesium (Mg), Zinc (Zn), Iron (Fe), Selenium (Se) Content of Flours and Breads

The analyses of Mg, Zn, Fe and Se were carried out by an external accredited laboratory of food analysis (UNI CEI EN ISO/IEC 17025:2005 [30], Accredia, Lab. n. 0490), using an inductively coupled plasma with mass spectrometer (ICP-MS) analytical method [31] (UNI EN 13805:2014). Results were expressed as mg/100 g for Mg, Zn and Fe and µg/100 g for Se on a dry weight basis.

### 2.6. Sample Extraction for Soluble and Insoluble Phenolic Compounds of Flours and Breads

Soluble and insoluble phenolic compounds were extracted from both flours and bread samples following the protocol proposed by Zaupa and colleagues [32]. The obtained extracts were dissolved in an opportune solvent and volume, used for the Ultra High-Performance Liquid Chromatography coupled to tandem mass spectrometry (UHPLC-MS/MS) analysis and the total phenolic content assay.

#### 2.6.1. Soluble and Insoluble Total Phenolic Content (TPC)

Soluble and insoluble total phenolic content (TPC) of bread samples were analysed by the *Folin–Ciocalteu’s* method [33]. A calibration curve using gallic acid as reference compound (100–1000 mg/Kg) was prepared for quantification. Results were reported as mg of gallic acid equivalents (GAE) per Kg on dry weight basis.

#### 2.6.2. Soluble and Insoluble Phenolic Acids Profile Using UHPLC-MS/MS

Phenolic acids (PA) profiling of bread samples was extracted according to Zaupa et al. [32] and analysed using a UHPLC Dionex Ultimate 3000 separation system coupled to a triple quadrupole mass spectrometer (TSQ Vantage; Thermo Fisher Scientific) following the protocol reported by Spaggiari and colleagues [34]. For quantification, two different calibration sets (0.05–5 and 5–100 mg/mL) were prepared using phenolic acids standard reference materials. Results were expressed as mg/Kg on dry weight basis.

### 2.7. Determination of Thiamine, Nicotinic Acid and Nicotinamide, and Folic Acid Content

For the extraction of the thiamine, nicotinic acid, nicotinamide and folic acid, the method proposed by Leporati et al. [35], was used. Results were expressed as mg/100 g with the only exception of folic acid (µg/100 g). The extracts were analysed using an Accela UPLC 1250 equipped with a linear ion trap MS (LTQ XL, Thermo Fisher Scientific Inc., San Jose, CA, USA) attached to an electrospray ionization probe (H-ESI-II; Thermo Fisher Scientific Inc., San Jose, CA, USA). Separation was performed on an Acquity UPLC HSS T3 (2.1 × 100 mm) column (Waters, Milford, MA, USA). The volume injected was 3 µL, and the oven temperature was set to 40 °C. The elution gradient was performed using CH_3_CN (0.1% formic acid) as mobile phase A and H_2_O (0.1% formic acid) as mobile phase B, at a flow rate of 0.3 mL/min, starting with 99% B and 1% A for 2 min, then eluent B decreased at 20% and A increased at 80% in 2 min, and maintained for further 2 min. Finally, the initial conditions were restored (total run time = 13 min). Data processing was performed using Xcalibur 2.2 software from Thermo Fisher Scientific Inc., (San Jose, CA, USA). The vitamins analysis was carried out in positive ionization mode, the MS worked with a capillary temperature set to 275 °C, while the source heater temperature was at 200 °C. The sheath gas (nitrogen) flow was 40 unit, while auxiliary and sweep gases (both nitrogen) were equal to 5 and 0 units, respectively. The spray voltage was 3.5 kV. The S-Lens value was 115 V. Vitamins were monitored using an MRM (multiple reaction monitoring) scan mode with the characteristic transitions reported in Appendix A.

### 2.8. Acceptability and Check-All-That-Apply (CATA) Analysis of Formulated Breads

Consumers’ sensory and hedonic perception of breads was assessed with an acceptability and CATA test. Breads were produced few hours prior to analysis following the recipe described above. After baking, the loaves were cooled and cut into half-slices of 1 cm thickness with a well-balanced crumb-to-crust ratio, packed separately in paper bags and labelled with a random three-digit code; the samples were simultaneously presented on a plate in randomized order and in blind condition. Water and unsalted crackers were provided as palate cleansers between samples. The panel consisted of 59 untrained consumers (46% male, 54% female, aged between 18 and 70 years old) who were asked to answer a CATA questionnaire consisting of 21 sensory characteristics listed in randomized order across assessors, selecting all the attributes they considered appropriate to describe the breads as well as their personal ‘ideal’ product. The terms used in the CATA test were the following: pleasant smell, unpleasant smell, smell of yoghurt, pleasant crust colour, unpleasant crust colour, golden crust colour, pale crust colour, soft crust, crunchy crust, pleasant crumb colour, unpleasant crumb colour, soft crumb, hard crumb, pleasant aftertaste, unpleasant aftertaste, salty taste, sweet taste, acid taste, mediocre, good, excellent bread.

After the CATA test, the consumers judged the acceptability of bread samples by rating aroma, taste, crust and crumb consistency, crust and crumb colour, appearance and overall acceptability with a 9-point hedonic scale [1 = dislike extremely, 2 = dislike very much, 3 = dislike, 4 = dislike slightly, 5 = neither like nor dislike, 6 = like slightly, 7 = like, 8 = like very much and 9 = like extremely].

### 2.9. Statistical Analysis

All analyses were performed at least in triplicate and reported as mean ± standard deviation (S.D. of each parameter are reported in Appendix A). To verify significant differences between samples, data obtained from the instrumental analyses and from the acceptability test were statistically analysed by performing one-way analysis of variance (ANOVA) followed by Duncan’s post-hoc test at <alpha> = 0.05 using SPSS Software Version 25.0 (SPSS Inc., USA). Data obtained from the CATA test were organized by compiling a contingency table to count how many times each attribute was used to describe each bread. Cochran’s Q statistic was performed to evaluate significant differences between products across the attributes. In order to identify relationships between attributes and samples, a sensory map of the products was obtained by performing a correspondence analysis (CA), performed with TIBCO Statistica Version 13.3 (TIBCO Software Inc., USA).

## 3. Results and Discussion

### 3.1. Milling and Technological Quality of Wheat

Table 1 shows the grain quality parameters of EPs in comparison to the cv. Bologna.

The protein content of all EPs grains (mean value ≈16.7%) was significantly (<alpha> = 0.05) higher than that of cv. Bologna (≈13%), although no differences were found among the EPs protein percentages. Protein content of cereal grains is an important parameter which determines their technological use [36], although protein levels have only partly a genetical basis and depend mostly on management practices and the environment [37]. The technological use of proteins is related to gluten proteins, i.e., glutenin and gliadin, located in the endosperm [36]. In general terms, there is a negative relationship between protein concentration and grain yield [37].

Furthermore, the alveographic parameters W and P/L are crucial for the assessment of wheat flours strength and extensibility [38]. The baking industry requires high W values (>180·10^−4^ J) combined with a balanced P/L index (0.40–0.50). As expected, significant differences (<alpha> = 0.05) were found between W parameter of EPs and cv. Bologna, which recorded the highest baking strength (288·10^−4^ J). Among EPs, ICARDA showed the highest W value (152.5·10^−4^ J), followed by BB (130.5·10^−4^ J) and BG (106.5·10^−4^ J). Besides, the P/L ratio showed a mean value ≈ 0.5 with no significant difference among samples (<alpha> = 0.05). Overall, ICARDA EP showed the most promising quality parameters among EPs for bread-making. Moreover, Bologna flour’s rheological parameters confirm its suitability for long-leavening bakery specialties [39].

During milling, to produce the same “Type” of flour (Type 1) according to the Italian legislation (Presidential Decree 187/2001), as defined by the ash level, middlings had to be added exclusively for FBo, indicating, for the Bologna variety, a different milling behaviour and/or an ash distribution particularly concentrated in the aleurone and bran layers, allowing for very high milling yields (i.e., ER at equal concentration of ashes) as already noticed by the Italian milling industry. In detail, the different milling behaviour of EPs compared to FBo can be attributed to a different grain hardness. Cv. Bologna is known—by industrial millers—to contain a small and hard kernel [39], and to have outstanding milling behaviour since the aleurone layer (with high ash content) detaches very well from the endosperm yielding a white flour with low ash content. On the contrary, from what we have observed in our study, milling of EPs caused portions of the aleurone layer to be released into the flour, resulting in higher ash levels.

### 3.2. Lipid Content and Fatty Acids Profile of Breads

Lipids play an important role on both sensory and technological quality of food products [40]. The crude fat analysed in bread was the highest for BBo and BG followed by BI and BB (Table 2).

Since the amount of extra virgin olive oil used in the recipe was the same, the differences could be attributed to the lipid content of the wheat grains. Concerning the fatty acids profile, results are reported as Appendix A. Oleic (C18:1), linoleic (C18:2), palmitic (C16:0) and linolenic (C18:3) acids were the most abundant in all breads formulated, in line with previous findings [41]. However, only C18:1 fatty acid was found significantly different between BI and BG, with the latter showing the highest content. In fact, the MUFA and PUFA varied among breads (Table 2), resulting in BG and BI with higher MUFA and BBo with higher PUFA. Endogenous wheat lipids have been studied to demonstrate their influence in breadmaking and showing their ability to stabilize the gas bubbles by aligning at gas-liquid interface during dough maturation [42]. In this context, the differences in both amount and quality of lipid fraction of the EPs, despite its lower content, might be considered a positive source of variation for producing breads. Moreover, the n-6/n-3 ratio is an important nutritional parameter, that shall stay below 1 [43]. All breads herein produced exhibited a healthy lipid index <1.

### 3.3. Total Dietary Fibre (TDF) and DF Classes of Flours and Breads

Dietary fibres are important components of cereal grains due to their well-documented beneficial properties [44]. The physiological effects are highly dependent on their physical and chemical characteristics (i.e., monomer composition, particle size, etc.) [45]. Regarding the TDF content of the breads herein formulated, no differences were found (Table 2). Overall, the breads might use the nutritional claim “source of fibre” (≥3 g TDF/100 g bread, [46]). Furthermore, a detailed analysis of the different dietary fibre classes of flours (Figure 2A) indicated a significantly higher HMWIDF content of cv. Bologna (<alpha> = 0.05) in respect to FI and FG. The latter could be ascribable to the middlings supplementation to native cv. Bologna flour which increased the final amount of these substances in flour. Concerning breads (Figure 2B) no differences were found among fibre classes. Moreover, the insoluble component slightly, although not significantly, increased after processing, ranging from 0.86 to 2.24 g/100 g in flours and from 1.60 to 2.40 g/100 g in breads. This could be related to the formation of resistant starch occurred during bread-making process (Figure 2C) [47,48]. In fact, the starch is subjected to gelatinization and retrogradation processes inducing physico-chemical modifications of available starch originally present in the flour. The derived component, resistant starch, is a fraction which results resistant to the digestion and contribute to increase the overall fibre fraction in breads. Likewise, lipids and dietary fibres greatly influence the bread dough rheological properties and its textural quality [49].

### 3.4. Selected Micronutrients Content of Flours and Breads

The content of important minerals (Mg, Zn, Fe, and Se) was analysed in flours (Table 3) and breads (Table 2).

The results obtained were in line with reference reported in various international databases [50,51,52]. Overall, the content of Mg, Zn, and Fe diminished after flour transformation, although Se content increased by around 3 times (as average) in all formulated breads. This reduction phenomenon could be related to some complexation in kneading and cooking phases [53] while the increased Se content was also reported elsewhere [54] probably due to the microorganism metabolism or the cell lysis itself. B-complex vitamins are important nutrients, essential for several human physiological functions. The content of thiamine, nicotinic acid and folic acid, were quantified in flours (Table 3) and then in breads (Table 2). Thiamine content in flours was higher for FBo and FG, in the range of values reported by Mihhalevski et al. [55]. While nicotinic acid was never detected in flours and breads, nicotinamide content increased significantly and among breads BBo totalized the highest content. The latter could be probably due to the fermentation of the sourdough processing [56]. Concerning the stability of B-vitamins during processing, thiamine can resist under the bread-making conditions (pH 4–5, high temperature), similar to nicotinamide [55]. Folic acid, was only found in FB samples, although after processing was detected as <LOQ. A high variability is usually found in group-B vitamin content of wheat grains, possibly due to the difficult analytical procedure and varietal differences [57].

### 3.5. Phenolic Compounds from Flours to Breads

Phenolic compounds in cereals are mainly present as simple phenolic acids, which are located in the outermost fraction of the seed (i.e., bran). For this reason, they can occur in soluble and mainly in insoluble form, thus strictly linked to the fibrous material of vegetable cells [58]. The TPC and PA profile of flours and formulated breads were reported in Table 4.

Concerning flour samples, the soluble component was negligible compared to the insoluble fraction. Furthermore, ferulic acid was the most abundant among PAs, as previously reported by other authors [34,58,59]. However, the TPC of bread showed a higher soluble component compared to the insoluble one. There are several potential explanations for this, mostly ascribed to the complex chemical reactions and modification involving metabolic processes and the high temperature during the transformation of flour into dough and bread. The most interesting phenolic acid transformation is ascribed to the action of fermentation which is shown to be crucial for the release of phenolic acids from the matrix, increasing their bioavailability for human digestion [60,61]. However, the thermal treatment applied during baking could be detrimental, degrading the thermolabile phenolic or complexing them in Maillard’s reaction-derived compounds lowering their final content in bread, as occurred in this study. Moreover, the formation of peptides which might interfere with the *Folin–Ciocalteu’s* assay and Maillard reaction’s soluble compounds [62] must be accounted for when interpreting the results of TPC method. In terms of abundance, both TPC and PA content of BBo were the highest among products, because of the middlings addition. However, variability in phenolic acid content of different wheat varieties is well known, with their biosynthesis strongly influenced by environmental stimuli [63]. As with other phenolic compounds, phenolic acids can act as antioxidants. Therefore, considering this property, a higher content of phenolic compounds might be translated to a higher protection against oxidation, hence a more stable product from both sensory and technological viewpoints.

### 3.6. Check-All-That-Apply (CATA) Analysis of Breads

Sensory analysis was carried out by including breads produced with cv. Bologna flour type 00 (without middlings addition, BBo t00 sample) as additional control. These breads were produced by applying the same processing conditions of other breads. The newly formulated breads were assessed by 59 consumers using CATA test. This method is valuable to understand the consumer perception of a specific food product. Therefore, a list of sensory attributes related to flavour, appearance, taste, texture and smell are evaluated by untrained panellists, which are allowed to select all the attributes better describing the product perception.

A preliminary correspondence analysis (CA) on the CATA dataset including BBo t00, bread produced using cv. Bologna flour type 00 and BBo t1, bread produced using cv. Bologna flour type 1 was performed (87% of the total variance explained, Appendix A). The breads prepared with EPs flours and the control sample BBo t1 were grouped in the centre of the plot, due to the fact that BBo t00 was perceived as a very different sample compared to the others (Appendix A). Data were confirmed by hedonic sensory evaluation data, which showed BBo t00 as the least appreciated sample (overall acceptability: 6.15 ± 1.20).

Since the aim of the work was to produce and characterize breads produced under the same processing conditions, assessing the suitability of EPs for bread-making in comparison with a commercial variety and from flours belonging to the same commercial type according to the Italian legislation (Type 1 flours), we repeated the correspondence analysis including only breads obtained from Type 1 flours. In such a way, the differences between EPs and the BBo t1 control could be better explained.

After executing a Cochran’s Q analysis of results, a significant difference (<alpha> = 0.05) in consumer perception for 14 out of 21 attributes among the different samples was found. In fact, assessors detected significant differences between samples for texture attributes (soft crust, crunchy crust, soft crumb), colour descriptors (unpleasant crust colour, pleasant and unpleasant crumb colour), smell (pleasant and unpleasant), taste (salty, acid), aftertaste (pleasant and unpleasant) and overall judgement (mediocre and excellent bread). Biplot shown in Figure 3 represents the visual configuration of the breads and their discriminating attributes in the first two dimensions of the correspondence analysis performed on the CATA dataset (92.9% of the total variance explained).

It can be observed that the ideal concept of bread for the panellists matched the “excellent” descriptor (right quadrant), and the breads prepared using EPs flours (BI, BG, BB) were grouped in the lower quadrants of Figure 3, and all intensely associated with sensory attributes of great impact for consumers. More in detail, BI was perceived by judges as having a “pleasant crumb colour”, “crunchy crust”, “soft crumb” and “pleasant smell”, attributes, which all had significant difference (<alpha> = 0.05) following Cochran’s Q test. In addition to previous positive sensory attributes, the judges perceived the presence of an “acid taste” and an “unpleasant aftertaste” for both BG and BB, which are clustered together and therefore closest to the attribute “mediocre bread”. Quality parameters such as bread volume, acidic taste and colour are deeply influenced by the sourdough processing due to enzymatic reactions occurring during fermentation [64]. However, the visual sensory characteristics referred to the crust colour (“unpleasant” and “pale”) did not have a significant difference. On the other hand, the control sample produced with cv. Bologna was visually located distant from the EPs-bread samples. According to CATA data, the bread was found close to “hard crumb”, “unpleasant smell”, “soft crumb” and “unpleasant crumb colour” descriptors, which could be related to the different flour preparation method affecting the sensory characteristics of the finished product [40]. Based on the frequency of the attribute selection, consumers described their ideal bread as having a pleasant crust and crumb colour (74% and 61%, respectively), pleasant smell (76%), golden crust (78%), crunchy crust (96%), soft crumb (87%), pleasant aftertaste (69%), salty taste (61%), and being good (31%) and excellent (63%). When comparing bread samples to the ideal product, no bread directly corresponded to the ideal one, but BI was the closest to it.

Overall, CATA test provided different sensory profiles descriptive of the bread samples, thus allowing an evaluation of the similarities and differences between breads produced by different types of flours.

The average scores obtained from hedonic sensory evaluation for each attribute of bread samples were reported in Table 5.

One-way ANOVA revealed significant differences (<alpha> = 0.05) between bread samples for crust texture, crumb colour and overall acceptability. More in detail, the crust texture of the breads made from EPs received significantly higher scores than the ones from the modern cv. Bologna (6.95, 6.85, 6.71 for BI, BG and BB, respectively). BG and BB were the preferred samples in terms of crumb colour (7.10 and 6.97, respectively), while BBo received the lowest score (6.39). In general, although overall acceptance was higher than 6 for all the breads, BI received the highest score (7.02), BG and BB had an intermediate score evaluation (6.75 and 6.73, respectively) and BBo resulted the least appreciated sample (6.36), thus integrating the results obtained with the CATA method.

## 4. Conclusions

The use of wheat (*Triticum aestivum.* L.) evolutionary populations cultivated in marginal areas under organic farming appeared to provide an environmental-friendly and market-oriented method to produce bread with an overall good nutritional quality (source of fibre) and final consumer perception. Moreover, this agricultural practice enhances the farmer’s expertise, allowing them to play a fundamental role in agrobiodiversity preservation. To the best of our knowledge, this is the first study on the overall quality and sensory attributes of novel breads formulated using wheat EPs cultivated in large scale, and finally compared to bread produced using a modern bread wheat variety. Although the technological quality for EP flours, as measured by the processing industry (W, P/L), seemed unsuitable for bread making, the sourdough baking carried out during the present study allowed excellent workability of the EPs doughs and good structure of the loaves with regular alveolation. From a chemical and nutritional perspective, the breads were comparable, despite middlings requiring addition for FBo to produce the same commercial “Type” of flour (Type 1). Considering consumer perception, which is an important parameter to accounted for in new product development, the bread produced using EPs was associated with positive sensory characteristics. Finally, the combination of sensory and chemical analysis permitted a better description of the utilization of wheat EPs for breadmaking. Results herein presented are valuable to pave the way for further studies dedicated to the formulation of new foodstuffs exploiting the EP potential in a strong collaboration with farmers.

## 5. Study Limitations and Future Perspectives

The quality of wheat is dependent on genotype but also on climatic conditions [32]. Since this study is based on one source (one year of wheat production), the outcomes of the research should be confirmed by analysing flours obtained from more sowing seasons. This is even more true in recent years where climate change is showing its effects. EPs have been shown to guarantee a stable production in a climate change scenario [7] and it would be interesting to evaluate whether they can also guarantee a stable grain technological quality.

Future perspectives should include the characterization of breads from a physicochemical and technological point of view.

## Figures and Tables

**Figure 1 foods-11-00495-f001:**
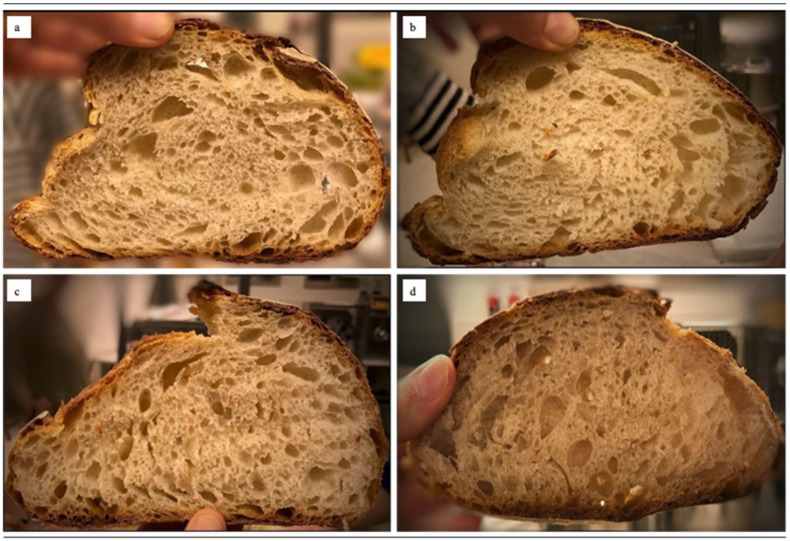
Pictures of the slices of the different breads. (**a**) BB, bread produced using Bio2 EP; (**b**) BI, bread produced using ICARDA EP; (**c**) BG, bread produced using Grossi EP; (**d**) BBo, bread produced using cv. Bologna.

**Figure 2 foods-11-00495-f002:**
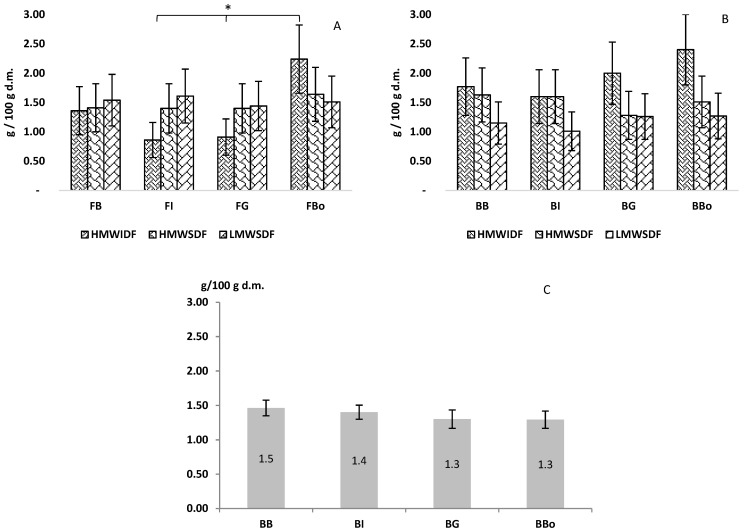
Classes of dietary fibres found in flours (**A**) and breads (**B**), together with resistant starch (**C**) determined in breads. Results are reported as mean ± standard deviation and expressed as g/100 g dry matter. * Indicates a significant difference <alpha> = 0.05. HMWIDF, high molecular weight insoluble dietary fibre; HMWSDF, high molecular weight soluble dietary fibre; LMWSDF, low molecular weight soluble dietary fibre; FB, BIO2 EP Type 1 flour; FI, ICARDA EP Type 1 flour; FG, Grossi EP Type 1 flour; FBo, cv. Bologna Type 1 flour; BB, bread produced using Bio2 EP; BI, bread produced using ICARDA EP; BG, bread produced using Grossi EP; BBo, bread produced using cv. Bologna.

**Figure 3 foods-11-00495-f003:**
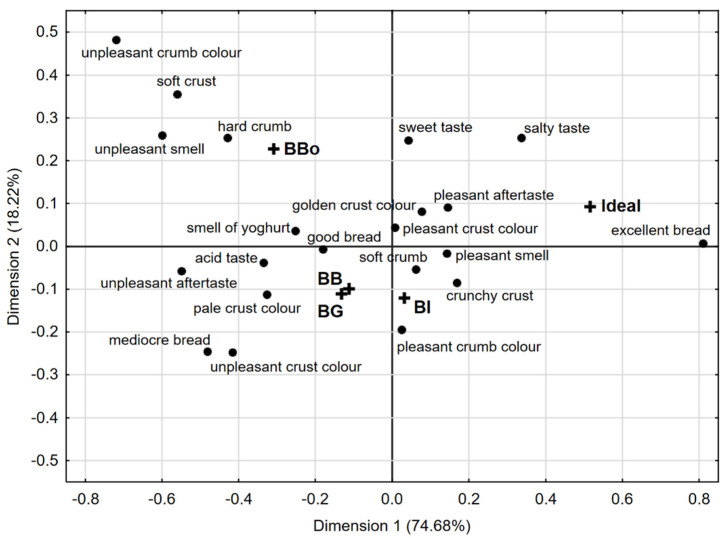
Correspondence analysis of the bread samples and sensory attributes. BB, bread produced using Bio2 EP; BI, bread produced using ICARDA EP; BG, bread produced using Grossi EP; BBo, bread produced using cv. Bologna.

**Table 1 foods-11-00495-t001:** Grain quality parameters of EPs and cv. Bologna.

Wheat	Test Weight (kg/hL)	Thousand Kernel Weight (g)	Protein Content (% d.m.)	Alveograph			
				W (10^−4^ J)	P (mm H_2_O)	L (mm)	P/L
Bio2 EP	74 ^a^	44 ^b^	16.82 ^b^	130.5 ^b^	67.5 ^d^	134.0 ^b^	0.5 ^a^
ICARDA EP	78 ^a^	45 ^b^	16.39 ^b^	152.5 ^c^	59.0 ^c^	129.5 ^b^	0.5 ^a^
Grossi EP	77 ^a^	47 ^c^	16.93 ^b^	106.5 ^a^	55.0 ^b^	108.5 ^a^	0.5 ^a^
Bologna	79 ^a^	32 ^a^	13.27 ^a^	288.0 ^d^	48.5 ^a^	98.5 ^a^	0.5 ^a^

Results are reported as mean (*n* = 3). Protein content is expressed as g/100 g on dry matter (d.m.). Standard deviation is reported in Appendix A. Different letters in the same column indicate significant differences among samples (<alpha> = 0.05). EP, evolutionary wheat population.

**Table 2 foods-11-00495-t002:** Nutritional and chemical composition of the bread formulated using the wheat evolutionary population (BB, BI and BG) and bread produced using flour from cv. Bologna wheat (BBo).

	BB	BI	BG	BBo
Energetic value (kJ) *	1005.0	1058.1	1041.1	961.6
Energetic value (kcal) *	240.2	252.9	248.8	229.8
Carbohydrates (g/100 g)	48.3 ^a^	49.7 ^a^	47.7 ^a^	46.2 ^a^
Total dietary fibre (g/100 g)	4.55 ^a^	4.22 ^a^	4.64 ^a^	5.18 ^b^
Lipids (g/100 g)	0.83 ^a^	1.0 ^b^	1.20 ^c^	1.22 ^c^
SFA (%)	31.8 ^a^	32.2 ^a^	31.7 ^a^	31.0 ^a^
MUFA (%)	42.9 ^a^	45.2 ^b^	45.2 ^b^	42.7 ^a^
PUFA (%)	25.3 ^c^	22.6 ^a^	23.0 ^b^	26.3 ^d^
Ω-6/Ω-9	0.53 ^b^	0.45 ^a^	0.45 ^a^	0.55 ^b^
Proteins (g/100 g)	12.4 ^a^	11.3 ^a^	12.1 ^a^	11.3 ^a^
Mg (mg/100 g)	24.5 ^a^	22.1 ^a^	24.1 ^a^	31.6 ^b^
Zn (mg/100 g)	0.85 ^b^	0.75 ^a^	0.82 ^a^	0.82 ^a^
Fe (mg/100 g)	1.37 ^c^	0.86 ^a^	1.09 ^b^	1.38 ^c^
Se (µg/100 g)	8.07 ^a^	8.11 ^a^	8.95 ^b^	8.77 ^b^
Thiamine (mg/100 g)	0.24 ^b^	0.18 ^a^	0.20 ^a^	0.43 ^c^
Nicotinic acid (mg/100 g)	<LOQ	<LOQ	<LOQ	<LOQ
Folic acid (µg/100 g)	<LOQ	<LOQ	<LOQ	<LOQ
Nicotinamide (mg/100 g)	1.77 ^b^	1.75 ^ab^	1.62 ^a^	2.18 ^c^

Results are reported as mean (*n = 3*). Standard deviation is reported in Appendix A. Different superscipts letters ^a–d^ in the same row indicate significant differences among samples (<alpha> = 0.05). <LOQ Folic acid: 5 µg/100 g; <LOQ Nicotinic acid: 0.01 mg/100 mg, Mg, magnesium; Zn, zinc; Fe, iron; Se, selenium; NAM, nicotinamide; BB, bread produced using BIO2 EP; BI, bread produced using ICARDA EP; BG, bread produced using Grossi EP; BBo, bread produced using cv. Bologna. *: Calories (kJ and kcal) were calculated as sum of nutritive components.

**Table 3 foods-11-00495-t003:** Micronutrients content in flours.

	FB	FI	FG	FBo
Mg (mg/100 g) *	29.2 ^a^	26.1 ^a^	28.5 ^a^	44.2 ^b^
Zn (mg/100 g) *	1.13 ^b^	0.97 ^a^	1.08 ^a^	1.17 ^b^
Fe (mg/100 g) *	1.80 ^c^	1.02 ^a^	1.29 ^b^	1.85 ^c^
Se (µg/100 g) **	2.66 ^a^	2.40 ^a^	3.39 ^b^	3.96 ^b^
Thiamine (mg/100 g) *	0.29 ^b^	0.22 ^a^	0.33 ^bc^	0.36 ^c^
Nicotinic acid (mg/100 g) *	<LOQ	<LOQ	<LOQ	<LOQ
Nicotinamide (mg/100 g) *	0.43 ^a^	<LOQ	0.51 ^ab^	0.56 ^b^
Folic acid (µg/100 g) **	21.8 ^b^	<LOQ ^a^	<LOQ ^a^	<LOQ ^a^

Results are expressed as mean (*n =* 3). Standard deviation is reported in Appendix A. Different superscripts letters ^a–c^ in the same column indicate significant difference (<alpha> = 0.05). * <LOQ, 0.01 mg/100 g; ** <LOQ, 0.5 µg/100 g. FB, BIO2 EP Type 1 flour; FI, ICARDA EP Type 1 flour; FG, Grossi EP Type 1 flour; FBo, cv. Bologna Type 1 flour.

**Table 4 foods-11-00495-t004:** Total phenolic content (TPC) and phenolic acid (PA) profile in their free (soluble) and bound (insoluble) forms.

Sample Flours	TPC		4-HB		p-C		Caff		t-Fer		c-Fer	Sin	
	Free	Bound	Free	Bound	Free	Bound	Free	Bound	Free	Bound		Free	Bound
	mg GAE/Kg d.m.		mg/Kg d. m.										
FB	256.22 ^a^	895.95 ^a^	0.07 ^a^	<LOQ	<LOQ	0.11 ^a^	0.21 ^b^	0.25 ^a^	1.53 ^b^	2.48 ^b^	0.59 ^a^	0.90 ^b^	1.25 ^b^
FI	217.70 ^a^	838.51 ^a^	0.09 ^a^	<LOQ	0.07 ^a^	0.12 ^a^	0.17 ^a^	0.25 ^a^	1.56 ^b^	2.21 ^a^	1.10 ^b^	0.80 ^a^	0.99 ^a^
FG	221.76 ^a^	912.84 ^a^	<LOQ	<LOQ	0.08 ^a^	0.12 ^a^	0.23 ^b^	0.32 ^b^	1.33 ^a^	3.25 ^c^	1.05 ^b^	0.96 ^b^	1.32 ^b^
FBo	390.49 ^b^	855.41 ^a^	<LOQ	<LOQ	<LOQ	0.18 ^b^	0.37 ^c^	0.52 ^c^	1.73 ^b^	2.66 ^ab^	0.91 ^b^	0.82 ^a^	1.17 ^b^
Breads													
BB	343.81 ^a^	217.97 ^a^	<LOQ	0.51 ^a^	<LOQ	1.47 ^b^	<LOQ	0.37 ^a^	1.32 ^a^	44.25 ^a^	21.47 ^a^	<LOQ	4.07 ^b^
BI	480.93 ^c^	182.89 ^a^	<LOQ	0.45 ^a^	<LOQ	1.21 ^a^	<LOQ	0.29 ^a^	1.86 ^a^	42.16 ^a^	15.96 ^a^	0.34 ^a^	3.73 ^a^
BG	414.83 ^b^	186.81 ^a^	<LOQ	0.73 ^b^	0.11 ^a^	1.79 ^c^	<LOQ	0.32 ^a^	2.29 ^ab^	47.23 ^a^	26.76 ^a^	0.38 ^a^	6.12 ^c^
BBo	490.29 ^c^	355.86 ^b^	0.21 ^a^	0.94 ^c^	0.11 ^a^	2.31 ^c^	<LOQ	0.63 ^b^	2.41 ^b^	73.64 ^b^	25.52 ^a^	<LOQ	6.19 ^c^

Results are expressed as mean (*n* = 3). Standard deviation is reported in Appendix A. Different superscripts letters ^a–c^ in the same column indicate significant difference (<alpha> = 0.05). <LOQ: 0.05 mg/kg. GAE, Gallic Acid Equivalents; d.m., dry weight; 4-HB, hydroxybenzoic acid; p-C, para coumaric acid; caff, caffeic acid; *t*-fer, *trans*-ferulic acid; *c*-fer, *cis*-ferulic acid; Sin, sinapic acid. FB, BIO2 EP Type 1 flour; FI, ICARDA EP Type 1 flour; FG, Grossi EP Type 1 flour; FBo, cv. Bologna Type 1 flour; BB, bread produced using BIO2 EP; BI, bread produced using ICARDA EP; BG, bread produced using Grossi EP; BBo, bread produced using cv. Bologna.

**Table 5 foods-11-00495-t005:** Sensory scores of breads obtained from acceptability test.

Bread	Texture		Colour		Appearance	Aroma	Taste	Overall Acceptability
	Crust	Crumb	Crust	Crumb				
BI	6.95 ^b^	7.05 ^a^	6.81 ^a^	6.78 ^ab^	7.05 ^a^	6.51 ^a^	6.69 ^a^	7.02 ^b^
BB	6.71 ^b^	6.92 ^a^	6.78 ^a^	6.97 ^b^	7.00 ^a^	6.46 ^a^	6.42 ^a^	6.73 ^ab^
BG	6.85 ^b^	6.78 ^a^	6.88 ^a^	7.10 ^b^	7.15 ^a^	6.27 ^a^	6.15 ^a^	6.75 ^ab^
BBo	6.08 ^a^	6.41 ^a^	6.83 ^a^	6.39 ^a^	6.59 ^a^	6.24 ^a^	6.08 ^a^	6.36 ^a^

Results are expressed as mean (*n =* 59). Standard deviation is reported in Appendix A. Different superscripts letters ^a–c^ in the same column indicate significant differences among samples (<alpha> = 0.05). BB, bread produced using BIO2 EP; BI, bread produced using ICARDA EP; BG, bread produced using Grossi EP; BBo, bread produced using cv Bologna.

## Data Availability

All data included in this study are available upon request by contacting the corresponding author.

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
