# Peer review of "Evolutionary Wheat Populations in High-Quality Breadmaking as a Tool to Preserve Agri-Food Biodiversity"

_foods, 2022, doi:10.3390/foods11040495_

Round 1
Reviewer 1 Report
The authors have summarized an interesting study on EP technological and sensory quality in sourdough breads. Indeed, literature is lacking in the area of biodiversity strategies in cereals production and the resulting technological performance. This is a step towards addressing that research gap.
I found that the manuscript will need a thorough copy edit for ESL grammatical/phrasing corrections. That notwithstanding, the current language does not impede the ability of the reader to understand what the authors are trying to communicate. It merely reads awkwardly at times. Because the editing needs in this area are significant, I have decided to allow the editorial team and the authors to address this issue rather than clouding my review with grammatical/phrasing corrections.
In terms of my questions to the authors and requests for revisions:
1) Abstract, Page 1, Lines 21-22: Revise to remove the sentence "These are also emerging trends...". This is not the focus of the paper. By removing this sentence, the next sentence will require revision to read "Wheat variety selection..."
2) Abstract, Page 1, Line 27: Revise to read "...with one bread made from a single commercial cultivar wheat flour..."
3) Abstract, Page 1, Line 28: The "B" in B-complex vitamins should be capitalized.
4) Introduction, Page 1, Line 41: The phrasing "conventional mainstream agriculture" is redundant. Remove "mainstream" to correct this redundancy.
5) Introduction, Page 1, Lines 42-43: Remove "ensuring a high yield and acceptable quality of the final product (i.e., kernels) but". This portion of the sentence is not necessary.
6) Introduction, Page 1, Line 42: The statement "...repeatedly as monocultures or in rotations that include only 2 species..." requires a citation, especially since it is not the norm in many regions of the world. For example, it is common for producers in North America to have a 4-year crop rotation consisting of 3 species with the 4th year devoted to fallow (i.e., a rest year for the soil). This doesn't include the agronomic management strategy of cover crops and other alternative cropping systems that are increasingly popular from a sustainability standpoint.
7) Introduction, Page 2, Lines 49-54: This paragraph is a logical place to introduce organic agriculture. It is noticeably absent in the introduction and requires some framing from the authors. A question to consider is whether there anything specific about EPs that requires or makes it easier to use organic agronomic practices.
8) Introduction, Page 2, Line 54: Remove the phrase "and its gastronomic identity". This phrase is a non-sequitur to the rest of the paragraph and irrelevant to the argument being pursued by the authors in this introductory section.
9) M&M, Page 3, Line 98: Define the specific organic agronomic practices used. There are many ways to farm organically. What is written is not specific enough.
10) M&M, Page 3, Line 119: Which model of Alveograph?
11) M&M, Page 4, Line 147: Why such a low RH? Typical RH conditions are ~80-85% RH.
12) M&M, Page 4, Lines 151-152: Which doughs required more water? Include the data in Table 1.
13) R&D, Page 6, Table 1: Please report P and L values separately in addition to the ratio. Reporting the individual P and L values gives context to the empirical rheological properties of each dough that is lost when reported in ratio format.
14) R&D, Page 6, Lines 259-261: Citation needed for the statement "...although protein levels have only partly a genetic basis and depend mostly on management practices and the environment". The authors also need to discuss the distribution of protein between the bran fraction and the endosperm. This is related to "technological use" as proteins commonly located in the bran fraction (primarily albumins/globulins) are not technologically functional.
15) R&D, Page 6, Lines 268-269: Again, this needs to be placed into context with the reporting of individual P and L values that create the P/L ratio. The P and L parameters influence which type of bread and breadmaking process is most suitable for a given sample, or if the sample is even suitable for bread production. Many North American soft wheats give P/L ratios of 0.50, but their individual P and L values show them to be highly unsuitable for bread production.
16) R&D, Page 6, Lines 270-275: This needs to be a separate paragraph. Additionally, the milling quality of the EPs needs to be discussed more thoroughly. Why do they mill differently than FBo? I assume its due to their kernel characteristics - they are probably smaller kernels relative to Bologna which leads to obvious deficiencies in the way they mill. However, the authors needs to explicitly state this and discuss it in context with their discussion of FBo.
17) R&D, Page 7, Lines 286-287: This sentence is true, but it would have been more technically correct to also evaluate the lipid profile of the raw flours for confirmation. It is not clear how much of the lipid profile was lost with the germ and bran fractions of the EP flours and how this compares to FBo where a significant amount of mids was added back to the white flour. I would recommend going back and testing the flours directly to compare with the breads to understand how processing transformed the lipid profiles.
18) R&D, Page 8, Lines 309-310: Regarding the statement "...the insoluble component increased after processing..." What happened to the more soluble fractions? Are they consumed during fermentation? Why are there differences among flours and not among breads? This discussion section needs to be expanded.
19) R&D, Page 8, Lines 313-315: The sentence beginning "Besides, dietary fibres..." should be removed. This statement does not add any value to the discussion of the results.
20) R&D, Page 9, Table 4: This table is awkwardly formatted on a portrait-oriented page. Please put it on its own page in a landscape orientation for easier reading.
21) R&D, Page 10, Lines 363-365: The statement "possibly ascribed to the formation of peptides which might interfere with the Folin-Ciocaltreu assay and Maillard reaction soluble compounds" does not take into account the role of fermentation. Fermentation is likely the primary route through which soluble TPC increase, not the formation of peptides and Maillard reaction soluble compounds. Fermentation is mentioned much later in the paragraph - far too late to contextualize this statement. The fermentation discussion needs to be moved to immediately precede this statement since it is the most likely mechanism influencing soluble TPC - the peptides and Maillard reaction soluble compounds can then be listed as potential but less likely causes.
22) R&D, Page 11, Lines 410-414: It would have been better to include a BBo without the mids component as a control in addition to the BBo with mids. The negative impact of bran on sensory perception is significant, and by not including a BBo without mids you are setting up a BBo with mids to be less preferred by default. Including a BBo without mids as another control would have likely influenced the CATA data as well as how the samples fell in the biplot. It is less likely that the EP breads would have fallen so close to "good bread" had a proper BBo control without mids been present in the sample set. I cannot see how the sensory results are valid without this additional BBo control without mids.
Overall, I recommend a major revision to the paper. The sensory results are essentially meaningless without a proper BBo control without mids. While the logic behind matching FBo to the commercial flour classification of the EP flours is sound, it doesn't negate the fact that, especially from a sensory standpoint, the BBo sample with mids was going to compare negatively to the EP breads by default.
Reviewer 2 Report
The manuscript deals of the use of evolutionnary and the evaluation populations of wheat for high quality breadmaking and their sensorial and chemical evaluation.
This is a original study which, I hope, will used as a example for other investigations both in evolutionnary populations but also for other wheat species.
The study is well structured, the used methods are sounds.
Some concerns remain and should be modified.
1- the abstract. In this part some numerical results should be added.
2- the quality of wheat is very dependent on genotype but also on climatic conditions. This information is lacking (even in supplementary file). Moreover This study is based on one source (one year of wheat production). It is important to be cautious in concluding on the results obtained.
3- figure S1 should be inserted in the main manuscript.
Reviewer 3 Report
The manuscript submitted by Folloni et al. mainly dealt with evolutionary wheat populations in high-quality breadmaking as a tool to preserve agro-food biodiversity. The manuscript is very practical and really important. However, the manuscript is hard to understand and poorly written. Some errors and problems need to be rectified as the following.
Abstract:
I didn't find the purpose of this study in the abstract, please shorten the background contents from line 19 to 25.
Introduction
I suggest you'd better introduce the CATA tests literature review about breads or flour products.
Materials and methods
line 96, I am confused with these names, please unify them or make it clear.
line 96 to 112, I think this part is not clear, and I can't understand them very clearly, please reorganize this part, especially those varieties names.
line 118, no literature citation here
line 119, (W, J 10-4) I can't understand the unit here.
line 129 and line 132, no citation here.
line 136, this equation is not clear, please redesign it, I suggest f, m and x can be used in lowercase, and you can introduce these parameters by an integrate symbol.
line 138 to 139, no definition of FBo, ER, FB, FG and FI
line 162, 168, 172 and 173, no literature citation here
line 178 and 180, no literature citation here
line 190, no citation here
line 209, H2O is not correct
line 227, 59 is not enough for CATA tests, please explain it.
Results and discussion
Table 1, why did Test weight have no significant differences mark.
Figure 1, A and B is very unclear, I suggest you'd better seperate HMWIDF, HMWSDF and LMWSDF, and mark the significance difference at the top of each bar.
Table 3, it's quite confused for me, why did you mark a,b behind Mg, Zn, Fe, Se and Thiamine, Nicotinic acid, Nicotinamide, and Folic acid, and you also mark a, b behind some data, and you also have a, b definition at the footnote, it was quite confused.
Table 5, why some data have significance mark, some data no mark
Conclusions
please add limitations and further research plan of this study.
Reviewer 4 Report
The manuscript entitled ‘Evolutionary wheat populations in high-quality breadmaking as a tool to preserve agro-food biodiversity’ is an analysis of the suitability of organic wheat flours obtained from evolutionary populations for a traditional sourdough bread-making process. The way of applying new products following the idea of mixing and sowing together as many wheat genotypes as possible to let the crop genetic adaptation is an interesting idea for a potential reader. The paper is nicely written in an understandable way but some minor improvements are needed to fulfill high standrads of Foods journal.
COMMENTS:
The English language should be carefully checked (should be ‘effect’ instead of ‘effects’ l. 23; ‘.as a matter …’ instead of ‘…as matter’ l. 77, etc)
Panelists should be trained before assessment. The explanation of statement in l. 227 is needed.
In all tables statistical significance should be presented as un upper index – this can be more readable for a reader
- 324 – 325 – text should be justified to the left
Table 4 is difficult to analyse. Standard deviation should be moved to supplementary material, only value and statistical significance should be presented in the main text.
‘<alpha>’ in l. 386 should be corrected
Statistical significance in table 5 should be completed (Texture crumb, Colour Crust, Appearance, Aroma, Taste) if there is no significant differences, the same letter should be placed.
Extended discussion should be devoted to the analysis presented in Fig 2. regarding the so called ‘ideal’ concept of bread and the analysis how to achieve ‘ideal’ bread with analyzed ingredients should be added.
The explanation why common commercial wheat bread was not chosen for analysis is needed.
Round 2
Reviewer 1 Report
This is a significant improvement over the original draft of the manuscript.
I would recommend acceptance once the authors address one final revision -
1) Please include a paragraph in the sensory section indicating that you did indeed run a CATA analysis with a BBO t00, and please include a short summary of the results (e.g., lines 206 - 213 of your response to reviewers). This will help contextualize the sensory results for readers who, like me, would like to understand how a control BBO t00 would compare to the BBO t1 and EP breads. I would also note the change in PC percentages (PC 1 = 61.99%, PC 2 = 25.71% with BBO t00 vs. PC 1 = 74.68%, PC 2 = 18.22% without BBO t00) in that summary.
Reviewer 3 Report
Accept
Author Response
Thank you.